# LFA-1: A potential key player in microglia-mediated neuroprotection against oxygen-glucose deprivation in vitro

Robin Jansen[1]*, Marc Pawlitzki[1‡], Michael Gliem[1‡], Sven G. Meuth[1‡], Stefanie Schreiber[2‡], Michael-W. Görtler[2‡], Jens Neumann[2]

**1** Medical Faculty and University Hospital Düsseldorf, Department of Neurology, Heinrich-Heine-University Düsseldorf, Düsseldorf, Germany, **2** Medical Faculty, Department of Neurology, Otto von Guericke University, Magdeburg, Germany

☯ These authors contributed equally to this work.
‡ MP, MG, SGM, SS and M-WG are also contributed also equally to this work
* robin.jansen@med.uni-duesseldorf.de

**Data Availability Statement:** All relevant data are within the manuscript and its Supporting Information files.

## Abstract

For the last 38 years, all neuroprotective agents for patients with ischemic stroke have failed in clinical trials. The innate immune system, particularly microglia, is a much-discussed target for neuroprotective agents. Promising results for neuroprotection by inhibition of integrins with drugs such as natalizumab in animal stroke models have not been translated into clinical practice. Our present study reveals the relevance of a β2 integrin, lymphocyte function-associated antigen-1 (LFA-1), as a potential key player in protecting neuronal cell death after oxygen-glucose deprivation in organotypic hippocampal cell cultures. In addition, we identified microglial cells as effector cells for LFA-1-mediated neuroprotection. The counterpart of LFA-1 on microglia is unclear, but we show strong expression of ICAM-5 in hippocampal neurons, suggesting a critical role for direct crosstalk between microglia and neurons for neuronal survival under oxygen-glucose deprivation. The enigma of neuroprotection after ischemic stroke remains to be solved, and our findings highlight the continuing importance and lack of understanding of integrin-mediated pathways after ischemic stroke and the need for further intensive research.

## Introduction

With 6.6 million deaths worldwide in 2019, stroke is the second leading cause of death right after ischemic heart disease [1]. Since 1985, when the efficacy of recombinant tissue plasminogen activator (rt-PA) was demonstrated in a rabbit stroke model, and despite promising experimental results of several neuroprotective agents for the acute and subacute phase of ischemic stroke in preclinical research, all agents have failed in clinical trials until to date [2,3]. One of the most discussed target to overcome the impressive „roadblock"of neuroprotective agents is the innate immune system. Ischemic events in the brain are accompanied by a complex inflammatory response of the innate immune system. Within this inflammatory response,

**Funding:** The author(s) received no specific funding for this work.

**Competing interests:** Michael Gliem: Speaking fees from Novartis and Pfizer. Mark Pawlitzki received honoraria for lecturing and travel expenses for attending meetings from Alexion, ArgenX, Bayer Health Care, Biogen, Hexal, Merck Serono, Novartis, Sanofi-Aventis and Teva. His research is funded by the by ArgenX, Biogen, Hexal and Novartis. Sven Meuth receives honoraria for lecturing, and travel expenses for attending meetings from Academy 2, Argenx, Alexion, Almirall, Amicus Therapeutics Germany, Bayer Health Care, Biogen, BioNtech, BMS, Celgene, Datamed, Demecan, Desitin, Diamed, Diaplan, DIU Dresden, DPmed, Gen Medicine and Healthcare products, Genzyme, Hexal AG, Impulze GmbH, Janssen Cilag, KW Medipoint, MedDay Pharmaceuticals, Merck Serono, Neuropoint, Novartis, Novo Nordisk, ONO Pharma, Oxford PharmaGenesis, Roche, Sanofi-Aventis, Springer Medizin Verlag, Chugai Pharma, QuintilesIMS, Teva, Wings for Life international and Xcenda. His research is funded by the German Ministry for Education and Research (BMBF), Bundesinstitut für Risikobewertung (BfR), Deutsche Forschungsgemeinschaft (DFG), Else Kröner Fresenius Foundation, Gemeinsamer Bundesausschuss (G-BA), German Academic Exchange Service, Hertie Foundation, Interdisciplinary Center for Clinical Studies (IZKF) Muenster, German Foundation Neurology and Alexion, Almirall, Amicus Therapeutics Germany, Biogen, Diamed, DGM e.V., Fresenius Medical Care, Genzyme, Gesellschaft von Freunden und Förderern der Heinrich-Heine-Universität Düsseldorf e.V., HERZ Burgdorf, Merck Serono, Novartis, ONO Pharma, Roche, and Teva. This does not alter our adherence to PLOS ONE policies on sharing data and materials

microglia act as a double-edged sword, exerting both protective and detrimental effects on angiogenesis, synaptic remodelling and neurogenesis as well as disrupting the blood-brain-barrier (BBB) [4]. A key to understanding the function of microglial cells lies in the variety of functional states, in the dynamics of invasion and in the respective mechanisms of communication with surrounding cells [4,5].

There are a number of known mechanisms of "microglial crosstalk" after ischemic stroke. for example, the crosstalk evolves from early neurotransmitter-induced activation of microglia triggered by neuronal death, followed by the release of neurotransmitters (e.g. IGF-1) with protective properties, cytotoxic nitric oxide (NO) and superoxide production, and finally to interleukin-1 (IL-1)-beta-induced activation of astroglial and oligodendrocyte cells [4,6–8]. However, several aspects of this microglial crosstalk remain incompletely understood. One of the most discussed interaction partners in microglial crosstalk is the β2-integrin, lymphocyte function-associated antigen-1 (LFA-1), which is expressed on activated microglia, but also on peripheral blood lymphocytes, neutrophils, monocytes, and natural killer cells [9–11]. Found throughout mammalian organisms, integrins are heterodimeric transmembrane glycoprotein receptors composed of an α- and β-subunit with the ability to signal inter- and intracellularly (inside-out and outside-in signalling) [12,13]. LFA-1 is a cell-surface receptor expressed on leukocytes and composed of CD11a (α-subunit) and CD18 (β-subunit), and finds its ligand in intracellular adhesion molecule 1–5 (ICAM) [9,14]. ICAMs are transmembrane glycoproteins composed of 2–9 immunoglobulin-like C2 domains and are expressed on the surface of neurons, leukocytes, endothelial cells and many cancer cells [15,16]. LFA-1 expression is mediated by a variety of cytokines such as interferon-gamma, IL-1 or tumor necrosis factor-alpha [15]. Activation of LFA-1 results in a two-way signal that mediates the prevention of apoptosis and cell proliferation [15].

Several lines of evidence support a role for LFA-1 in the pathogenesis of multiple sclerosis, Parkinson's disease and ischemic stroke [9,16,17]. Pharmacological inhibition of integrins is an emerging therapy and has already led to a number of well-established treatments for various diseases, such as the αIIbβ3 inhibitor tirofiban (coronary heart disease or stroke), the α4β7 antibody vedolizumab (ulcerative colitis) or the α4β1 inhibitor natalizumab (multiple sclerosis) [18]. For the latter, promising experimental results in reducing hemispheric ischemic neuronal damage could not be translated to humans, leading to a failed double-blind phase IIb clinical trial in 2020 [12,19]. Reasons for the failure include timing or interactions with other medications such as aspirin or rtPA as well as a small numbers of patients as well [20].

In addition to LFA-1, inhibition of another integrin, the very late antigen-4 (VLA-4), which is composed of CD49b (α-subunit) and CD29 (β-subunit), has shown clear neuroprotective properties in animal studies, although this effect has not been consistent across different stroke models [21,22]. Together with the serological evidence of reduced T-cell expression of CD11a (the α-subunit of LFA-1) in multiple sclerosis patients receiving natalizumab therapy, this raises with the above mentioned experimental evidence of reduced hemispheric damage in natalizumab-treated animals, this raises the question of the role of integrins, and in particular LFA-1, in ischemic cell damage [23]. Our previous findings demonstrate that microglial neuroprotection is dependent on CD11 expression on microglia and, microglia cell functionality is one fundamental mechanism underlying LFA-1-mediated neuroprotection [24].

In the present study, we demonstrate in organotypic hippocampal slice cultures that specific inhibition of microglial LFA-1 by the compound BIRT377 results in a dose-dependent increase in neuronal death following oxygen-glucose deprivation. BIRT377 specifically interacts with LFA-1 by non-covalently binding to the CD11a chain, preventing LFA-1 from binding to its ligands [25,26]. In addition, inhibition of LFA-1 abolished the neuroprotective effect of externally applied microglia (BV-2). The counterpart of microglial LFA-1 remains unclear,

but it is tempting to speculate that ICAM-5 may be the neuronal part due to its strong expression in hippocampal neurons.

## Results

### Microglia engage in cell-cell contact with neurons after OGD

We have previously shown that interference with microglial CD11a, a component of LFA-1 (CD11a/CD18), exacerbates neuronal cell death, presumably by inhibiting microglial migration to vulnerable neurons, but this has not been further investigated. First, we tested whether the previously observed physical interaction between neurons and exogenously applied microglia, termed as "capping", also exists between neurons and endogenous microglia. Therefore, we used organotypic hippocampal slice cultures from transgenic mice expressing Thy-1-eYPF in a subset of pyramidal neurons, specifically in the CA-1 area of the hippocampus. Endogenous microglia were labelled with rhodamine-labelled IB-4. We performed two-photon microscopy (TPM) of the slice culture 24 hours after oxygen-glucose deprivation and confirmed the "capping" under these endogenous conditions (Fig 1A). The molecular mechanism of this physical interaction under ischemic conditions is unknown. In addition, immunohistochemistry revealed a strong expression of ICAM-5 on THY1-expressing pyramidal neurons in the CA-1 region of the hippocampus (Fig 1B), whereas neurons did not express ICAM-1 under normal conditions (Fig 1C).

### LFA-1 antagonism by BIRT377 dose-dependently increases neuronal cell death after OGD

The next step was to investigate whether inhibition of LFA-1 function with BIRT377 had an effect on neuronal survival after OGD. The duration of the OGD was 30 minutes and resulted in submaximal neuronal cell death, allowing us to also detect exacerbation of neuronal cell death [27]. As there are few microglia present in hippocampal slice cultures, with the majority on the surface, the effect of endogenous microglia per se was unclear [28].

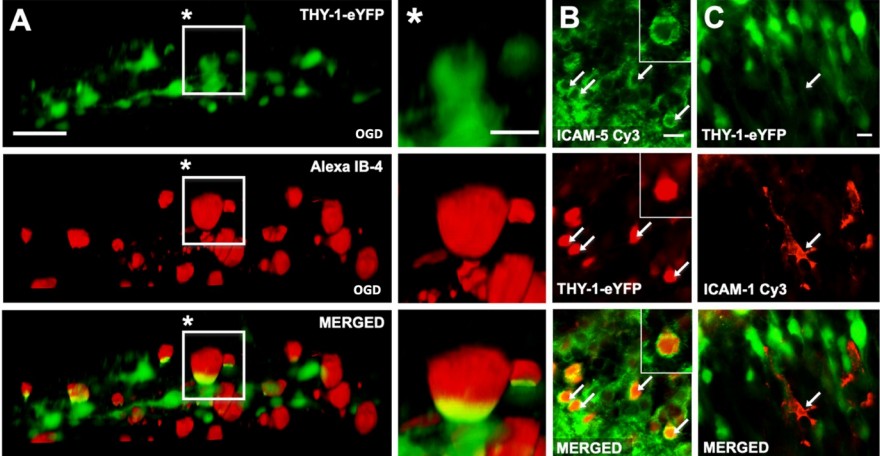

**Fig 1. After OGD, neurons express the ICAM-5 protein and establish direct cell-cell interactions with microglia.** A: THY1-expessing Neurons are co-localized with IB-4 positive microglia. B: Hippocampal neurons express ICAM-5 (shown as false color display for contrast reasons). C: Hippocampal neurons were negative for ICAM-1, ICAM-1 positive cells are according to their morphology microglial cells. Scale bar: A 30 μm, A* 15 μm, B 20 μm, C, 15 μm.

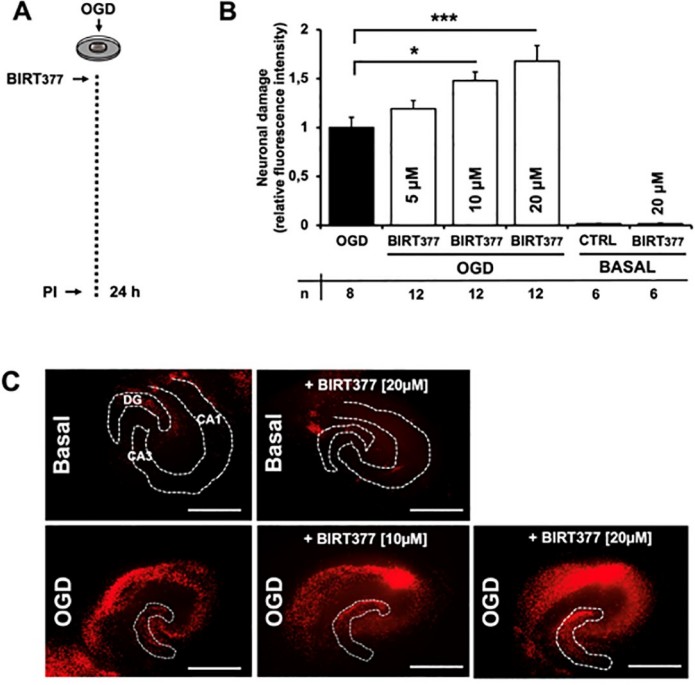

**Fig 2. After OGD preconditioning, LFA-1 antagonism by BIRT377 increases neuronal cell death in a dose-dependent manner.** A: Within one hour after oxygen-glucose-deprivation, BIRT377 was applied with consecutive cell death measurement after twenty-four hours via propium iodid incorporation (PI). B: After twenty-four hours, quantification of neuronal death in hippocampal CA1–3 region was determined by PI incorporation after application of 5, 10 or 20 μM BIRT377 after OGD and basal treatment (* $p < 0,05$, ***$p < 0.001$ vs OGD).n = 26 Error bars indicate SEM. C: Compared to basal conditions, BIRT377 application dose-dependently increases neuronal cell death with accentuation in CA1-regions (representative PI fluorescent images). Dotted lines and annotations (DG = dentate gyrus, CA = cornu ammonis) illustrate anatomical structure of the hippocampus. CA-region in OGD treated slices is not encircled for increased recognition of the PI positive cells in the CA-1 region. Scale bars: **C** 1 mm.

LFA-1 antagonism significantly exacerbated neuronal cell death after OGD in a dose-dependent manner (Fig 2B). Fig 2C shows representative fluorescence images of the densito-metric quantification (Fig 2C). The data suggest that LFA-1 function is critical for neuronal survival after ischemic brain injury.

## LFA-1 receptor function is crucial for exogenous microglia to mediate neuroprotection after OGD

Previous work with antisense CD11a treatment resulted in impaired microglial migration to vulnerable neurons, thereby interfering with microglial-mediated neuroprotection [24]. Microglia no longer migrated into the slice culture but remained on the top layer. With BIRT377, we were able to block LFA-1 function at any time after OGD, allowing us to investigate whether microglia that have had time to migrate into the slice are still capable of exerting neuroprotective effects after blocking LFA-1 function at later time points. Previous data showed that the majority of microglia had migrated into the slice culture at 6 hours after OGD. Therefore, we blocked LFA-1 at 1 and 6 hours after OGD (Fig 3A). While application of LFA-1-blocking BIRT377 at 1 hour after OGD abolished the previously observed microglia-mediated neuroprotection, application at 6 hours showed significantly less interference with neuroprotection (Fig 3B). The present results indicate an essential role for sufficient LFA-1

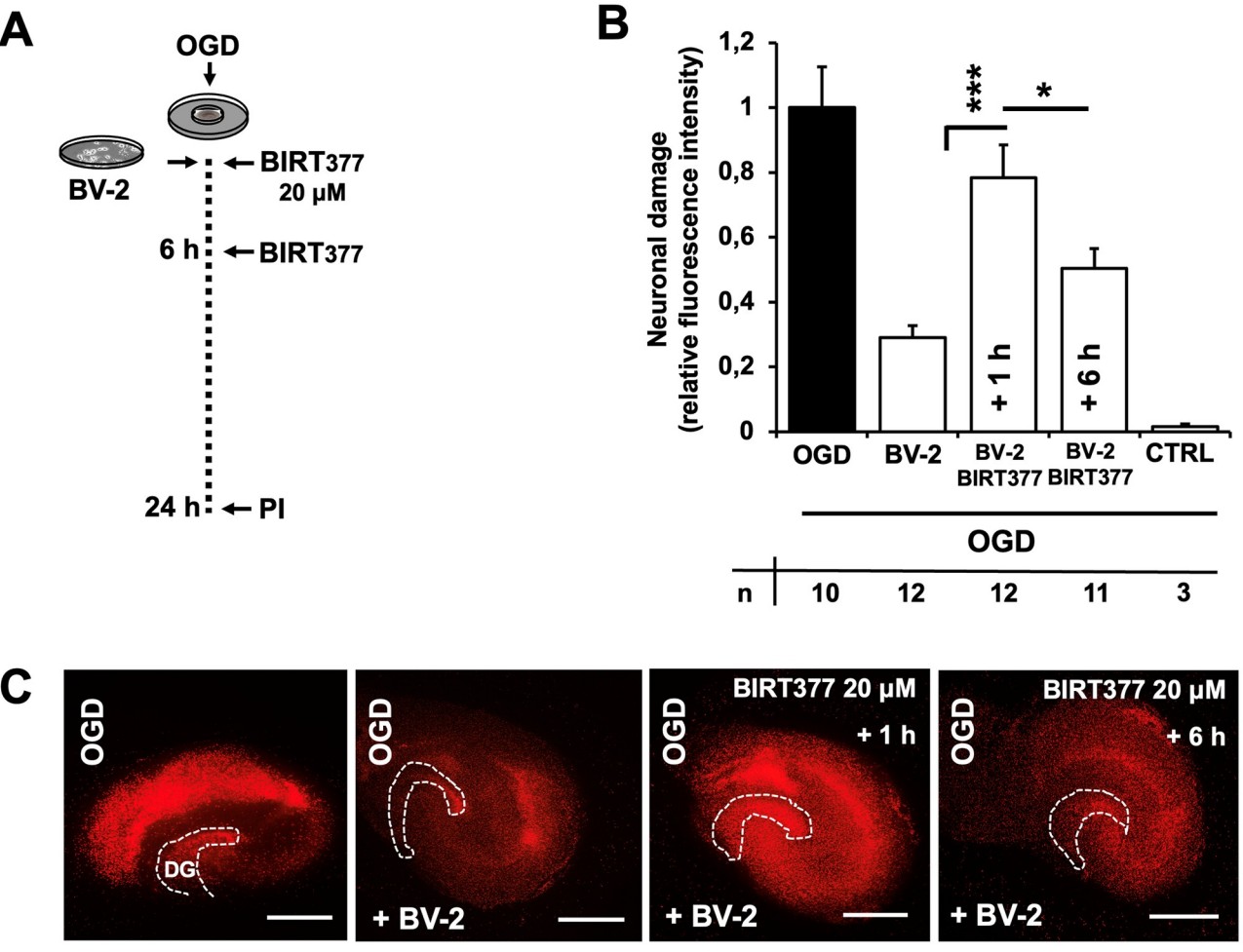

**Fig 3. Neuroprotective properties of microglia are mediated by LFA-1 receptor function in a time dependent manner after OGD.** A. Following the external addition of BV-2 microglia, 20 μM BIRT377 was applied either immediately after or six hours after oxygen-glucose deprivation, with cell death measured by PI after twenty-four hours. B: After twenty-four hours, quantification of neuronal death in hippocampal CA1–3 region was determined by PI incorporation after OGD alone, with externally applied BV-2 microglia and BIRT377 after one and six hours (* p < 0,05, ***p < 0.001 vs OGD) n = 27. Error bars indicate SEM. C: Representative PI fluorescence images of hippocampal slices used in B. Scale bars: **C** 1 mm.

receptor function on microglia and hence neuroprotection, but also that disruption of LFA-1 signalling only partially inhibits the neuroprotective properties of microglia, implicating a role for other signalling pathways and possibly time points.

## Discussion

Since all previous immunological stroke therapies have shown efficacy only in animal models and have failed to translate to humans, this study makes an important contribution to elucidating the mechanisms of microglial crosstalk after cerebral ischemia and defines potential new therapeutic targets [29].

The importance of microglial-driven neuroinflammation after ischemic stroke for infarct size and clinical outcome has been demonstrated in numerous animal studies. Thus, drug intervention in the sense of a mediator role to influence microglial activity in the direction of neuroprotection and repair seems obvious [13].

The aim of this study was to demonstrate the importance of LFA-1 receptor function for neuronal survival in oxygen-glucose-depleted organotypic hippocampal slices and to evaluate the role of LFA-1 in microglial neuroprotection. The present in vitro conditions allowed us to look in isolation at neuronal survival in OHCs under ischemic conditions under LFA-1 receptor blockade after one and six hours with and without additional microglial activity. If we assume a cell-to-cell interaction between microglia and neurons, the first requirement is evidence of spatial proximity. The present TPM and immunofluorescence staining shows clustering of neuronal and microglial cells as well as expression of the LFA-1 ligands ICAM-1 and ICAM-5 (Fig 1A–1C). This proximity allows microglia to interact with surrounding neurons via numerous surface receptors such as the lipopolysaccharide receptor CD14, CD36, the purinergic receptor P2Y6 or toll-like receptors by releasing a variety of inflammatory cytokines such as IL-1-beta, tumor necrosis factor-alpha, interferon-gamma or nitric oxide synthase (iNOS) and protective neurotrophins such as NGF, BDNF and NT-3 [30]. To investigate the role of LFA-1-receptor function, we were able to establish a dose-dependent relationship between neuronal cell death and LFA-1 receptor blockade after OGD (Fig 2A–2C). BIRT377 blocks LFA-1(CD11a), which is critical for the migration of exogenously added microglia into the slice [28]. Our ex vivo studies have shown that the migration of exogenous microglia into the slices is essential for exerting neuroprotection [28,31]. In addition, other studies have shown that ablation of endogenous microglia in vivo exacerbates neuronal damage following experimental stroke. Both observations suggest that both endogenous and exogenous microglia are neuroprotective rather than destructive in stroke models [32].

Since cell invasion does not occur in the present in vitro model (in which LFA-1 receptor function plays an integral role), the cause of the exacerbation of neuronal damage must therefore lie in a disruption of existing cell-to-cell communication by OGD [33]. Since our previous work showed a dependence of microglial neuroprotection on CD11 expression, it has been suggested that the key to LFA-1 mediated neuroprotection lies in microglial cell function [28]. To address the role of LFA-1-mediated, time-dependent microglial neuroprotection, we externally applied BV-2 microglia on cell culture with subsequent BIRT377 application after one and six hours. The expression of LFA-1 on BV-2 microglia is well known and has been described as early as 1996 by Hailer and colleagues, as well as in Pateau et al. (2017) and was confirmed by us by antisense CD11a (LFA1) knockdown in our previous work [28,34,35].

Time-dependent microglial functional states have been known for many years. As microglial activity moves along a continuum between a cytotoxic M1 response and a reparative M2 response, therefore an investigation of LFA-1 receptor function as a function of time was performed. Interestingly, our results showed a greater effect on neuronal cell death after blocking microglial activity at six hours than at one hour (Fig 3A–3C). This provides evidence that early microglia-neuron interactions are less determinant of neuronal cell death than late ongoing cell-to-cell communication. It is important to note that our model does not take in to account a monocytic cell migration and does not consider mechanisms and signaling pathways that could only come into play as a result of the increased cell numbers associated with invasion. Adding the results of the transcriptome data from Xue et al., which revealed a continuum between cytotoxic and reparative function rather than strict dichotomous function, it can also be assumed that in the present model we have only considered a specific functional state of microglial cells induced by oxygen-glucose deprivation [36].

Since we showen that microglia a frequently "cap" neurons and postulate this phenomenon as a neuroprotective property, it is interesting to identify the molecular mechanism behind this physical interaction. LFA-1 may be crucial for migration through neuronal tissue but may also serve as an interaction with neurons. Common counterparts of LFA-1 are ICAMs. Neurons only show ICAM-1 positivity in regions of the dentate gyrus where neurogenesis occurs,

whereas hippocampal neurons were positive for ICAM-5. The interaction between ICAM-5 and LFA-1 has been demonstrated in an in vitro assay with neurons and T cells [16]. It is still unclear whether the interaction between LFA-1 and ICAM-5 is responsible for capping. Further experiments interfering with ICAM-5, e. g. with antibodies, are warranted to clarify the physical interaction between microglia and neurons under ischemic conditions and whether ICAM-5 is crucial to induce or elicit neuroprotection.

In conclusion, our results provide important insights into the importance of LFA-1 receptor function in the context of neuronal survival after oxygen-glucose deprivation. This work should provide the impetus for further studies in animal models according to the STAIR guidelines of different stroke mechanisms such as tMCAO or photothrombosis, gender and comorbidities, as well as in existing LFA-1 knock-out models to gain further insight into the LFA-1 receptor and validation to enable potential translation into a clinical trial.

## Material und methods

### Oxygen–glucose deprivation

Oxygen-glucose deprivation (OGD) was performed as previously described [24]. To establish experimental conditions, sterile six-well culture plates (TPP) were prepared by saturating them with a 5% $CO_2$/95% $N_2$ atmosphere for 10 minutes. Membrane inserts containing a maximum of three outer hair cells (OHCs) were then placed in 1 ml of glucose-free medium in the culture plates. The OHCs were then subjected to oxygen-glucose deprivation (OGD) by being exposed to a hypoxic chamber (Billups-Rothenberg) with no glucose medium and a $N_2$/$CO_2$ atmosphere for either 30 or 40 minutes. After the OGD period, the OHCs were returned to normal conditions. Reference control cultures were maintained in normal medium with glucose under normoxic conditions. The cultures were then analysed 24 hours after the OGD treatment. BIRT377 was kindly provided by Terence Kelly (Boehringer Ingelheim).

### Analysis of cell death

At 24 hours after oxygen-glucose deprivation (OGD), cell death was assessed by measuring the incorporation of propidium iodide (PI) into the cells. The method for analysing cell death was performed as previously described [24]. Specifically, cultures were exposed to medium containing PI (10 μmol) for 2 hours at a temperature of 33˚C. Fluorescence images were acquired semi-automatically (Nikon motorized stage; LUCIA software) and analysed by densitometry to quantify necrotic cell death (LUCIA Image analysis software). The region of interest for analysis was defined based on transmission light images, specifically targeting the CA area (CA1-3) and excluding the dentate gyrus. A control square of 150 × 150 μm, located in the stratum moleculare outside the pyramidal cell layer, was used for automatic background, correction. To consolidate data from different experiments, the densitometric mean of each insult within a single experiment was normalised to 1 and expressed as relative fluorescence intensity. All additional data were presented as relative fluorescence intensity, reflecting the extent of damage caused by the insult.

### Organotypic hippocampal slice cultures

To reduce the suffering of the animals on death, rapid decapitation was performed after anaesthesia with isoflurane and in the absence of a pain response. Hippocampal interface organotypic cultures were prepared from postnatal day 7–9 Wistar rats (Harlan Winkelmann) as previously described [24]. For immunofluorescence microscopy experiments and two photon microscopy imaging, hippocampal slice cultures were prepared from transgenic B6.Cg-TgN

(Thy1-YFP)16Jrs mice (The Jackson Laboratory; distributed by Charles River), which express enhanced yellow fluorescent protein (EYFP) at high levels in subsets of central neurons, including the pyramidal cells of the hippocampus. These mice are genetically modified to express enhanced yellow fluorescent protein (EYFP) at high levels in specific populations of central neurons, including hippocampal pyramidal cells. Hippocampi were dissected and sliced transversely at a thickness of 350 μm (The Mickle Laboratory Engineering). Slices were transferred to Millicell membranes (Millipore). Cultures were maintained at 37˚C in 1 ml of serum-based medium containing 50% MEM-Hanks, 25% HBSS, 17 mm HEPES, 5 mm glucose, pH 7.8 (Cell Concepts), 1 mm l-glutamine (Biochrom), 25% horse serum (Invitrogen), and 0.5% gentamycin (Biochrom) for 2–3 d. Cultures were then maintained in serum-free medium (Neurobasal A medium with B27 complement, 5 mm glucose, 1 mm l-glutamine). The OHCs were selected by adding a non-toxic concentration of propidium iodide (PI) (2 μg/ml; Sigma-Aldrich) 12 h before the experimental start. PI-negative slices were considered to be healthy, and only these slices were used for the experiments. LFA-1 function was inhibited by application of the compound BIRT377 (Tocris), BIRT377 were applied in different concentration to native OHCs and in another set of experiments after addition of exogenous microglia on OHCs at different time points.

## Application of BV-2 microglia onto OHCs

Isolated BV-2 microglia were trypsinated (trypsin/EDTA; Biochrom), centrifuged at 500 × g for 2 min, and finally resuspended in Neurobasal medium (Invitrogen). As indicated, OHCs were fixed with 4% paraformaldehyde (PFA) and mounted with 3:1 PBS/glycerol. OHCs were then further analyzed with the indicated microscopy approach.

## Culture of microglia cell line BV2

The BV2 microglia were cultured in (D)MEM supplemented with 10% fetal calf serum (FCS) (Biochrom), 1% pen/strep (Biochrom), and 1% L-glutamine (Biochrom) at a density not exceeding $5 \times 10^5$ cells/ml and maintained in 5% CO2 at 37˚C. For the application of BV2 microglia, we trypsinated cells (trypsin/EDTA) (Biochrom) and then centrifuged them at 500g for 5 min and resuspended them in neurobasal medium (Gibco). Cell concentration was determined by counting cells in a "Neubauer" hemocytometer, and viability was assessed by trypan blue staining (0.4% trypan blue in PBS) (Sigma). BV2 microglia were applied directly onto 10-day-old OHSC in a volume of 2 μl of neurobasal medium containing $8 \times 10^4$ BV2 microglial cells, resulting in pathophysiologically relevant number of microglia numbers in the slice that closely resembling those found in vivo after global ischemia.

## Two-photon microscopy

For two-photon microscopy, the endogenous microglial cells were stained with Alexa 568-conjugated Griffonia simplicifolia isolectin B4 (Invitrogen). OHCs were subjected to two-photon microscopy (TPM) using an Olympus BX51WI stage equipped with a XLUMPL ×20, NA 0.95 water dipping lens and a multibeam scan head (LaVision Biotech, Bielefeld, Germany) run at 16 beams with full laser power. Image detection was done with a cooled CCD camera (Imager Intense, LaVision, Goettingen, Germany). Brain slices were imaged in live modus at 800 and 920 nm wavelength of the laser (MaiTai, Spectra physics) using a scanning window of $90 \times 150$ μm. These positions defined the maximum cube of tissue available for imaging. This cube was then scanned with a resolution in Z of 0.5 μm, first at 920 nm and then at 800 nm wavelength without a filter. The emission of EYFP at 800 nm was negligible as was the emission of Alexa 568 at 920 nm. Image stacks were exported as two independent 8-Bit multilayer

TIFF stacks and subsequently reconstructed using the AMIRA software package (Berlin, Germany).

## Animals

As mentioned above, for organotypic cultures Wistar rats ((Harlan Winkelmann GmbH, Borchen, Germany) were sacrificed on day 7–9 postnatal. For immunofluorescence microscopy experiments transgenic B6. Cg-TgN (Thy1-YFP)16Jrs mice (The Jackson Laboratory, Charles River, Wilmington, MA) were sacrificed. Briefly, 7–9 day old Wistar rats (Harlan Winkelmann GmbH, Borchen, Germany) were decapitated and their brains quickly removed under sterile conditions. After isolation of the hippocampi, their dorsal halves were cut transversely at 350 μm using a McIlwain tissue chopper (The Mickle Laboratory Engineering Co., Guildford, UK).

## Statistical analysis

All data are given as mean ± SEM. Statistical analysis was performed by one-way ANOVA followed by *post hoc* comparison (Tukey's test). A value of $p < 0.05$ was considered statistically significant.

## Author Contributions

**Conceptualization:** Marc Pawlitzki, Jens Neumann.

**Data curation:** Robin Jansen, Marc Pawlitzki, Jens Neumann.

**Formal analysis:** Jens Neumann.

**Investigation:** Jens Neumann.

**Methodology:** Jens Neumann.

**Project administration:** Jens Neumann.

**Supervision:** Marc Pawlitzki.

**Validation:** Marc Pawlitzki.

**Visualization:** Robin Jansen, Jens Neumann.

**Writing – original draft:** Robin Jansen, Marc Pawlitzki, Jens Neumann.

**Writing – review & editing:** Robin Jansen, Marc Pawlitzki, Michael Gliem, Sven G. Meuth, Stefanie Schreiber, Michael-W. Görtler, Jens Neumann.

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
