## [Decision Letter · Decision Letter 0]

7 Jun 2024

PONE-D-24-13219LFA-1: A potential key player in microglia-mediated neuroprotection against oxygen-glucose deprivation in vitroPLOS ONE

Dear Dr. Jansen,

Thank you for submitting your manuscript to PLOS ONE. After careful consideration, we feel that it has merit but does not fully meet PLOS ONE’s publication criteria as it currently stands. Therefore, we invite you to submit a revised version of the manuscript that addresses all the points raised during the review process.

We look forward to receiving your revised manuscript.

Kind regards,

Mária A. Deli, M.D., Ph.D.

Academic Editor

PLOS ONE

Journal Requirements:

"Michael Gliem: Speaking fees from Novartis and Pfizer. 

Mark Pawlitzki  received honoraria for lecturing and travel expenses for attending meetings from Alexion, ArgenX, Bayer Health Care, Biogen, Hexal, Merck Serono, Novartis, Sanofi-Aventis and Teva. His research is funded by the by ArgenX, Biogen, Hexal and Novartis. 

Sven Meuth receives honoraria for lecturing, and travel expenses for attending meetings from Academy 2, Argenx, Alexion, Almirall, Amicus Therapeutics Germany, Bayer Health Care, Biogen, BioNtech, BMS, Celgene, Datamed, Demecan, Desitin, Diamed, Diaplan, DIU Dresden, DPmed, Gen Medicine and Healthcare products, Genzyme, Hexal AG, Impulze GmbH, Janssen Cilag, KW Medipoint, MedDay Pharmaceuticals, Merck Serono, Neuropoint, Novartis, Novo Nordisk, ONO Pharma, Oxford PharmaGenesis, Roche, Sanofi-Aventis, Springer Medizin Verlag, Chugai Pharma, QuintilesIMS ,Teva, Wings for Life international and Xcenda.

His research is funded by the German Ministry for Education and Research (BMBF), Bundesinstitut für Risikobewertung (BfR), Deutsche Forschungsgemeinschaft (DFG), Else Kröner Fresenius Foundation, Gemeinsamer Bundesausschuss (G-BA), German Academic Exchange Service, Hertie Foundation, Interdisciplinary Center for Clinical Studies (IZKF) Muenster, German Foundation Neurology and Alexion,  Almirall, Amicus Therapeutics Germany, Biogen, Diamed, DGM e.v., Fresenius Medical Care, Genzyme, Gesellschaft von Freunden und Förderern der Heinrich-Heine-Universität Düsseldorf e.V., HERZ Burgdorf, Merck Serono, Novartis, ONO Pharma, Roche, and Teva."

5. We note that your Data Availability Statement is currently as follows: All relevant data are within the manuscript and its Supporting Information files.

6. Please amend the manuscript submission data (via Edit Submission) to include author Dr. Michael-W. Görtler.

7. We note that you have included the phrase “data not shown” in your manuscript. Unfortunately, this does not meet our data sharing requirements. PLOS does not permit references to inaccessible data. We require that authors provide all relevant data within the paper, Supporting Information files, or in an acceptable, public repository. Please add a citation to support this phrase or upload the data that corresponds with these findings to a stable repository (such as Figshare or Dryad) and provide and URLs, DOIs, or accession numbers that may be used to access these data. Or, if the data are not a core part of the research being presented in your study, we ask that you remove the phrase that refers to these data.

**Additional Editor Comments:**

The manuscript was evaluated by two experts. They suggested some important amendments that are needed for acceptance including changes in the images, improvements of the text. At least one further experimental figure showing the mechanism of the observed neuroprotective effect is needed.==============================

Reviewers' comments:

Reviewer's Responses to Questions

**Comments to the Author**

1. Is the manuscript technically sound, and do the data support the conclusions?

Reviewer #1: Partly

Reviewer #2: Partly

2. Has the statistical analysis been performed appropriately and rigorously? 

Reviewer #1: Yes

Reviewer #2: Yes

3. Have the authors made all data underlying the findings in their manuscript fully available?

Reviewer #1: Yes

Reviewer #2: Yes

4. Is the manuscript presented in an intelligible fashion and written in standard English?

Reviewer #1: Yes

Reviewer #2: Yes

5. Review Comments to the Author

Reviewer #1: This study shows the critical role of the β2 integrin, LFA-1, in microglia-mediated neuroprotection during oxygen-glucose deprivation (OGD) in hippocampal cell cultures. The research demonstrates that LFA-1 is necessary for neuronal survival under ischemic conditions, as its inhibition with the compound BIRT377 resulted in a dose-dependent increase in neuronal death. ICAM-5, which is strongly expressed in hippocampal neurons, was identified as a potential ligand for LFA-1, indicating an important interaction between microglia and neurons that is essential for neuroprotection. These findings highlight the significance of LFA-1 in neuroprotective signaling and emphasize the need for further investigation into integrin-mediated pathways in the context of ischemic stroke.

Minor concerns:

In the introduction, it is mentioned that IGF-1 is an example of a neurotransmitter, but its canonical function is as a growth factor mainly released from the liver, although also produced in many tissues. IGF-1's main activity is to induce proliferation; however, in the brain, it has been related to survival and neuroprotection.

What happens to the BV-2 microglial cells that are co-incubated with the hippocampal organotypic cultures? Is there any kind of tissue penetration? Do microglial BV-2 cells secrete proinflammatory cytokines? Are they M1 or M2 phenotype? Are they pre-treated to express a specific phenotype?

Did you try higher doses of BIRT377?

Is PI incorporation specific to neuronal cells?

Please include better pictures of the OHCs in Figures 2C and 3C, as they are very small, and without nuclei counterstaining, it is very hard to identify the damage zone and the affected cells.

Authors show that N=7. How was this calculated? What is the effect size? Was it calculated a priori?

Does BIRT377 have a differential effect on endogenous microglia and BV-2?

Major concerns:

The abstract is very general and does not provide important information such as methodology, a summary of results, and a brief conclusion. It also states that there is no single therapy to induce neuroprotection after a stroke in patients, which is hard to believe. I recommend being more cautious with statements like this.

The control in Figures 2C and 3D show a strong discrepancy in propidium iodide incorporation intensity. The timeline indicates that they are exposed to PI for 24 hours; if there is any other difference in the protocol that explains this fluorescence inconsistency, it should be included in the manuscript.

I strongly recommend including a conclusion or summary at the end of the discussion, which includes only technical English, avoiding expressions such as “Like a stone into water.” This can be confusing for some international readers.

Stressed microglia are reported to promote an M1 proinflammatory phenotype, which can be harmful to neural tissue. Why does BIRT377, once blocking OGD in non-stressed BV2, increase survival (Figure 2B)? Also, endogenous microglia and BV2 are likely to have opposing responses since one is stressed and the other is not. Please include a discussion about this.

The number of results seems a little short. It would be interesting to see what the mediators are in the effect exogenous microglia upon survival in hippocampal tissue cultures. Maybe qPCR or ELISA for cytokines and neurotrophins could complement this work.

Reviewer #2: In this manuscript, the authors investigated LFA-1 inhibition with BIRT377 on neuronal death after oxygen-glucose deprivation (OGD) in cultured hippocampal slices. The authors found that BIRT377 treatment enhanced neuronal death after OGD, and abolished the neuroprotective effect of transplanted BV2 cells on OGD-preconditioned slices. Below are some comments on the manuscript.

(1) In figure 1C, the authors show expression of ICAM5 but not ICAM1 in neurons. At which experimental conditions (OGD vs. Normal) was this done?

(2) In figure 3, the authors show that BV2 cell transplantation is neuroprotective on OGD-preconditioned slices, which can be partially abolished by LFA-1 inhibition. The expression of LFA-1 in BV2 cells should be experimentally confirmed.

(3) There is a lack of scale bars on the immunofluorescent images in all figures.

(4) It is difficult to see the morphology of the slices in figure 2C and figure 3C. The authors should either provide bright-light images of the slices together with the fluorescent images or enhance the intensity of the fluorescent images to show the slice morphology.

(5) The manuscript needs careful proof-reading: in several occasions, the authors start the sentence with “previously”, “previous studies” or “previous data” without providing any literature reference; in line 7 of the abstract: “hippocampal cell culture” should be “hippocampal slice culture”; in line 16 of the introduction, IGF-1 is not a neurotransmitter; in the “Organotypic hippocampal slice cultures” section of the method: repeated texts in lines 5-8; for microglia labelling: it is stated in the main text that rhodamine-IB4 was used; however, in the method section it is indicated that Alexa568-IB4 was used.

6. PLOS authors have the option to publish the peer review history of their article (what does this mean?). If published, this will include your full peer review and any attached files.

Reviewer #1: No

Reviewer #2: No

---

## [Author Response · Author response to Decision Letter 0]

30 Jul 2024

Response to Reviewers

Reviewer #1: This study shows the critical role of the β2 integrin, LFA-1, in microglia-mediated neuroprotection during oxygen-glucose deprivation (OGD) in hippocampal cell cultures. The research demonstrates that LFA-1 is necessary for neuronal survival under ischemic conditions, as its inhibition with the compound BIRT377 resulted in a dose-dependent increase in neuronal death. ICAM-5, which is strongly expressed in hippocampal neurons, was identified as a potential ligand for LFA-1, indicating an important interaction between microglia and neurons that is essential for neuroprotection. These findings highlight the significance of LFA-1 in neuroprotective signaling and emphasize the need for further investigation into integrin-mediated pathways in the context of ischemic stroke.

Minor concerns:

In the introduction, it is mentioned that IGF-1 is an example of a neurotransmitter, but its canonical function is as a growth factor mainly released from the liver, although also produced in many tissues. IGF-1's main activity is to induce proliferation; however, in the brain, it has been related to survival and neuroprotection.

What happens to the BV-2 microglial cells that are co-incubated with the hippocampal organotypic cultures? Is there any kind of tissue penetration?

Answer: Using two-photon microscopy, we demonstrated that added BV-2 microglia penetrates the slice in a time-dependent manner following a OGD [1]

Do microglial BV-2 cells secrete proinflammatory cytokines? Are they M1 or M2 phenotype? 

Answer: Measuring cytokines was not within the scope of this study; therefore, we did not include it. While we have not determined the microglia phenotype, we assume that they are initially M1 polarized.

Are they pre-treated to express a specific phenotype?

Answer: No, the BV-2 microglia were not pre-treated. 

Did you try higher doses of BIRT377?

Answer: No, we did not apply higher doses. 

Is PI incorporation specific to neuronal cells?

Answer: No, PI intercalates into the DNA of every necrotic cell. 

Please include better pictures of the OHCs in Figures 2C and 3C, as they are very small, and without nuclei counterstaining, it is very hard to identify the damage zone and the affected cells.

Answer: PI is a nuclear counterstain. 

Authors show that N=7. How was this calculated? What is the effect size? Was it calculated a priori?

Answer: We need to apologize. The number of slices were false. We corrected the numbers (higher) in the manuscript.

Does BIRT377 have a differential effect on endogenous microglia and BV-2?

Answer: We did not differentiate between the two types. However, the data indicate that BIRT377 had an effect in both conditions: ODG with endogenous microglia (Figure 2) and exogenous microglia (Figure 3). This suggests that BIRT377 acts similarly on both types of microglia.

Major concerns:

The abstract is very general and does not provide important information such as methodology, a summary of results, and a brief conclusion. It also states that there is no single therapy to induce neuroprotection after a stroke in patients, which is hard to believe. I recommend being more cautious with statements like this.

The control in Figures 2C and 3D show a strong discrepancy in propidium iodide incorporation intensity. The timeline indicates that they are exposed to PI for 24 hours; if there is any other difference in the protocol that explains this fluorescence inconsistency, it should be included in the manuscript.

I strongly recommend including a conclusion or summary at the end of the discussion, which includes only technical English, avoiding expressions such as “Like a stone into water.” This can be confusing for some international readers.

Answer: We adapted the manuscript accordingly. 

Stressed microglia are reported to promote an M1 proinflammatory phenotype, which can be harmful to neural tissue. Why does BIRT377, once blocking OGD in non-stressed BV2, increase survival (Figure 2B)? Also, endogenous microglia and BV2 are likely to have opposing responses since one is stressed and the other is not. Please include a discussion about this.

Answer: BIRT377 blocks LFA-1(CD11a) which is crucial for migration of externally added microglia into the slice. [1] Our ex vivo studies demonstrated that the migration of exogenous microglia into the slice is essential for exerting neuroprotection [1,2]. Additionally, other studies have shown that the ablation of endogenous microglia in vivo exacerbates neuronal damage following experimental stroke. Both observations indicate that both endogenous and exogenous microglia are neuroprotective rather than destructive in stroke models [3]Both observations indicate that endogenous, as well as exogenous microglia are neuroprotective rather than destructive in stroke models. These aspects are included in the discussion.

The number of results seems a little short. It would be interesting to see what the mediators are in the effect exogenous microglia upon survival in hippocampal tissue cultures. Maybe qPCR or ELISA for cytokines and neurotrophins could complement this work.

Answer: We regret that the project has been terminated. Future studies will aim to delve deeper into the underlying mechanisms; we hope we can conduct such study in short term. 

Reviewer #2: In this manuscript, the authors investigated LFA-1 inhibition with BIRT377 on neuronal death after oxygen-glucose deprivation (OGD) in cultured hippocampal slices. The authors found that BIRT377 treatment enhanced neuronal death after OGD, and abolished the neuroprotective effect of transplanted BV2 cells on OGD-preconditioned slices. Below are some comments on the manuscript.

(1) In figure 1C, the authors show expression of ICAM5 but not ICAM1 in neurons. At which experimental conditions (OGD vs. Normal) was this done?

Answer: It was done under normal conditions.

(2) In figure 3, the authors show that BV2 cell transplantation is neuroprotective on OGD-preconditioned slices, which can be partially abolished by LFA-1 inhibition. The expression of LFA-1 in BV2 cells should be experimentally confirmed.

Answer: The expression of LFA-1 has been documented in other publications [4,5]. We conducted CD11a (LFA-1) antisense knockdown in BV-2 microglia, which resulted in impaired microglial migration [1]. This information has been included in the manuscript.

(3) There is a lack of scale bars on the immunofluorescent images in all figures.

Answer: We added the scale bars. 

(4) It is difficult to see the morphology of the slices in figure 2C and figure 3C. The authors should either provide bright-light images of the slices together with the fluorescent images or enhance the intensity of the fluorescent images to show the slice morphology.

Answer: We enhanced the quality of images. 

(5) The manuscript needs careful proof-reading: in several occasions, the authors start the sentence with “previously”, “previous studies” or “previous data” without providing any literature reference; in line 7 of the abstract: “hippocampal cell culture” should be “hippocampal slice culture”; in line 16 of the introduction, IGF-1 is not a neurotransmitter; in the “Organotypic hippocampal slice cultures” section of the method: repeated texts in lines 5-8; for microglia labelling: it is stated in the main text that rhodamine-IB4 was used; however, in the method section it is indicated that Alexa568-IB4 was used.

Answer: We completed proofreading and made changes according to your addressed points.

References:

1. Neumann J, Gunzer M, Gutzeit HO, Ullrich O, Reymann KG, Dinkel K. Microglia provide neuroprotection after ischemia. FASEB Journal. 2006;20. doi:10.1096/fj.05-4882fje

2. Mitrasinovic OM. Microglia Overexpressing the Macrophage Colony-Stimulating Factor Receptor Are Neuroprotective in a Microglial-Hippocampal Organotypic Coculture System. Journal of Neuroscience. 2005;25: 4442–4451. doi:10.1523/JNEUROSCI.0514-05.2005

3. Lalancette-Hebert M, Gowing G, Simard A, Weng YC, Kriz J. Selective Ablation of Proliferating Microglial Cells Exacerbates Ischemic Injury in the Brain. Journal of Neuroscience. 2007;27: 2596–2605. doi:10.1523/JNEUROSCI.5360-06.2007

4. Paetau S, Rolova T, Ning L, Gahmberg CG. Neuronal ICAM-5 inhibits microglia adhesion and phagocytosis and promotes an anti-inflammatory response in LPS stimulated microglia. Front Mol Neurosci. 2017;10. doi:10.3389/fnmol.2017.00431

5. Hailer NP, J T JI, Nitsch R. Resting Microglial Cells In Vitro: Analysis of Morphology and Adhesion Molecule Expression in Organotypic Hippocampal Slice Cultures. Glia. 1996.

---

## [Decision Letter · Decision Letter 1]

9 Sep 2024

PONE-D-24-13219R1LFA-1: A potential key player in microglia-mediated neuroprotection against oxygen-glucose deprivation in vitroPLOS ONE

Dear Dr. Jansen,

Thank you for submitting your manuscript to PLOS ONE. After careful consideration, we feel that it has merit but does not fully meet PLOS ONE’s publication criteria as it currently stands. Therefore, we invite you to submit a revised version of the manuscript that addresses the points raised during the review process.

The manuscript was improved, but there are still problems to be solved. The attached clean copy is not the same as the version with tracked changes. The authors need to address ALL comments of both reviewers and these should be incorporated in the revised text as well. I suggest the authors to use color highlight in yellow in revision 2 as compared to the original submission to ensure all changes are visible and clear. 

We look forward to receiving your revised manuscript.

Kind regards,

Mária A. Deli, M.D., Ph.D.

Academic Editor

PLOS ONE

Journal Requirements:

Reviewers' comments:

Reviewer's Responses to Questions

**Comments to the Author**

1. If the authors have adequately addressed your comments raised in a previous round of review and you feel that this manuscript is now acceptable for publication, you may indicate that here to bypass the “Comments to the Author” section, enter your conflict of interest statement in the “Confidential to Editor” section, and submit your "Accept" recommendation.

Reviewer #1: All comments have been addressed

Reviewer #2: (No Response)

2. Is the manuscript technically sound, and do the data support the conclusions?

Reviewer #1: Yes

Reviewer #2: (No Response)

3. Has the statistical analysis been performed appropriately and rigorously? 

Reviewer #1: Yes

Reviewer #2: (No Response)

4. Have the authors made all data underlying the findings in their manuscript fully available?

Reviewer #1: Yes

Reviewer #2: (No Response)

5. Is the manuscript presented in an intelligible fashion and written in standard English?

Reviewer #1: Yes

Reviewer #2: (No Response)

6. Review Comments to the Author

Reviewer #1: Authors addressed all the concerns, however, I still think the amount of results is a little short for a journal as PLOS ONE

Reviewer #2: The manuscript has not been revised at all in the clean version, except that one or two sentences were inserted in the method section. For the version with tracked changes, some revisions have been made based on reviewers' comments. However, not all comments had been addressed even though the authors claimed that they did so in the responses to reviewers.

7. PLOS authors have the option to publish the peer review history of their article (what does this mean?). If published, this will include your full peer review and any attached files.

Reviewer #1: No

Reviewer #2: No

---

## [Author Response · Author response to Decision Letter 1]

11 Oct 2024

Response to Reviewers

Dear academic editors, dear reviewers,

Please find attached the updated manuscript and a version in track change mode with the changes highlighted in yellow as discussed. You will also find the answers to the reviewers' questions below. All answers are included in the uploaded manuscript files.

Reviewer #1: This study shows the critical role of the β2 integrin, LFA-1, in microglia-mediated neuroprotection during oxygen-glucose deprivation (OGD) in hippocampal cell cultures. The research demonstrates that LFA-1 is necessary for neuronal survival under ischemic conditions, as its inhibition with the compound BIRT377 resulted in a dose-dependent increase in neuronal death. ICAM-5, which is strongly expressed in hippocampal neurons, was identified as a potential ligand for LFA-1, indicating an important interaction between microglia and neurons that is essential for neuroprotection. These findings highlight the significance of LFA-1 in neuroprotective signaling and emphasize the need for further investigation into integrin-mediated pathways in the context of ischemic stroke.

Minor concerns:

In the introduction, it is mentioned that IGF-1 is an example of a neurotransmitter, but its canonical function is as a growth factor mainly released from the liver, although also produced in many tissues. IGF-1's main activity is to induce proliferation; however, in the brain, it has been related to survival and neuroprotection.

What happens to the BV-2 microglial cells that are co-incubated with the hippocampal organotypic cultures? Is there any kind of tissue penetration?

Answer: Using two-photon microscopy, we demonstrated that added BV-2 microglia penetrates the slice in a time-dependent manner following a OGD [1]

Do microglial BV-2 cells secrete proinflammatory cytokines? Are they M1 or M2 phenotype? 

Answer: Measuring cytokines was not within the scope of this study; therefore, we did not include it. While we have not determined the microglia phenotype, we assume that they are initially M1 polarized.

Are they pre-treated to express a specific phenotype?

Answer: No, the BV-2 microglia were not pre-treated. 

Did you try higher doses of BIRT377?

Answer: No, we did not apply higher doses. 

Is PI incorporation specific to neuronal cells?

Answer: No, PI intercalates into the DNA of every necrotic cell. 

Please include better pictures of the OHCs in Figures 2C and 3C, as they are very small, and without nuclei counterstaining, it is very hard to identify the damage zone and the affected cells.

Answer: PI is a nuclear counterstain. 

Authors show that N=7. How was this calculated? What is the effect size? Was it calculated a priori?

Answer: We need to apologize. The number of slices were false. We corrected the numbers (higher) in the manuscript.

Does BIRT377 have a differential effect on endogenous microglia and BV-2?

Answer: We did not differentiate between the two types. However, the data indicate that BIRT377 had an effect in both conditions: ODG with endogenous microglia (Figure 2) and exogenous microglia (Figure 3). This suggests that BIRT377 acts similarly on both types of microglia.

Major concerns:

The abstract is very general and does not provide important information such as methodology, a summary of results, and a brief conclusion. It also states that there is no single therapy to induce neuroprotection after a stroke in patients, which is hard to believe. I recommend being more cautious with statements like this.

The control in Figures 2C and 3D show a strong discrepancy in propidium iodide incorporation intensity. The timeline indicates that they are exposed to PI for 24 hours; if there is any other difference in the protocol that explains this fluorescence inconsistency, it should be included in the manuscript.

I strongly recommend including a conclusion or summary at the end of the discussion, which includes only technical English, avoiding expressions such as “Like a stone into water.” This can be confusing for some international readers.

Answer: We adapted the manuscript accordingly. 

Stressed microglia are reported to promote an M1 proinflammatory phenotype, which can be harmful to neural tissue. Why does BIRT377, once blocking OGD in non-stressed BV2, increase survival (Figure 2B)? Also, endogenous microglia and BV2 are likely to have opposing responses since one is stressed and the other is not. Please include a discussion about this.

Answer: BIRT377 blocks LFA-1(CD11a) which is crucial for migration of externally added microglia into the slice. [1] Our ex vivo studies demonstrated that the migration of exogenous microglia into the slice is essential for exerting neuroprotection [1,2]. Additionally, other studies have shown that the ablation of endogenous microglia in vivo exacerbates neuronal damage following experimental stroke. Both observations indicate that both endogenous and exogenous microglia are neuroprotective rather than destructive in stroke models [3]Both observations indicate that endogenous, as well as exogenous microglia are neuroprotective rather than destructive in stroke models. These aspects are included in the discussion.

The number of results seems a little short. It would be interesting to see what the mediators are in the effect exogenous microglia upon survival in hippocampal tissue cultures. Maybe qPCR or ELISA for cytokines and neurotrophins could complement this work.

Answer: We regret that the project has been terminated. Future studies will aim to delve deeper into the underlying mechanisms; we hope we can conduct such study in short term. 

Reviewer #2: In this manuscript, the authors investigated LFA-1 inhibition with BIRT377 on neuronal death after oxygen-glucose deprivation (OGD) in cultured hippocampal slices. The authors found that BIRT377 treatment enhanced neuronal death after OGD, and abolished the neuroprotective effect of transplanted BV2 cells on OGD-preconditioned slices. Below are some comments on the manuscript.

(1) In figure 1C, the authors show expression of ICAM5 but not ICAM1 in neurons. At which experimental conditions (OGD vs. Normal) was this done?

Answer: It was done under normal conditions.

(2) In figure 3, the authors show that BV2 cell transplantation is neuroprotective on OGD-preconditioned slices, which can be partially abolished by LFA-1 inhibition. The expression of LFA-1 in BV2 cells should be experimentally confirmed.

Answer: The expression of LFA-1 has been documented in other publications [4,5]. We conducted CD11a (LFA-1) antisense knockdown in BV-2 microglia, which resulted in impaired microglial migration [1]. This information has been included in the manuscript.

(3) There is a lack of scale bars on the immunofluorescent images in all figures.

Answer: We added the scale bars. 

(4) It is difficult to see the morphology of the slices in figure 2C and figure 3C. The authors should either provide bright-light images of the slices together with the fluorescent images or enhance the intensity of the fluorescent images to show the slice morphology.

Answer: We enhanced the quality of images. 

(5) The manuscript needs careful proof-reading: in several occasions, the authors start the sentence with “previously”, “previous studies” or “previous data” without providing any literature reference; in line 7 of the abstract: “hippocampal cell culture” should be “hippocampal slice culture”; in line 16 of the introduction, IGF-1 is not a neurotransmitter; in the “Organotypic hippocampal slice cultures” section of the method: repeated texts in lines 5-8; for microglia labelling: it is stated in the main text that rhodamine-IB4 was used; however, in the method section it is indicated that Alexa568-IB4 was used.

Answer: We completed proofreading and made changes according to your addressed points.

References:

1. Neumann J, Gunzer M, Gutzeit HO, Ullrich O, Reymann KG, Dinkel K. Microglia provide neuroprotection after ischemia. FASEB Journal. 2006;20. doi:10.1096/fj.05-4882fje

2. Mitrasinovic OM. Microglia Overexpressing the Macrophage Colony-Stimulating Factor Receptor Are Neuroprotective in a Microglial-Hippocampal Organotypic Coculture System. Journal of Neuroscience. 2005;25: 4442–4451. doi:10.1523/JNEUROSCI.0514-05.2005

3. Lalancette-Hebert M, Gowing G, Simard A, Weng YC, Kriz J. Selective Ablation of Proliferating Microglial Cells Exacerbates Ischemic Injury in the Brain. Journal of Neuroscience. 2007;27: 2596–2605. doi:10.1523/JNEUROSCI.5360-06.2007

4. Paetau S, Rolova T, Ning L, Gahmberg CG. Neuronal ICAM-5 inhibits microglia adhesion and phagocytosis and promotes an anti-inflammatory response in LPS stimulated microglia. Front Mol Neurosci. 2017;10. doi:10.3389/fnmol.2017.00431

5. Hailer NP, J T JI, Nitsch R. Resting Microglial Cells In Vitro: Analysis of Morphology and Adhesion Molecule Expression in Organotypic Hippocampal Slice Cultures. Glia. 1996.

---

## [Decision Letter · Decision Letter 2]

5 Nov 2024

LFA-1: A potential key player in microglia-mediated neuroprotection against oxygen-glucose deprivation in vitro

PONE-D-24-13219R2

Dear Dr. Jansen,

We’re pleased to inform you that your manuscript has been judged scientifically suitable for publication and will be formally accepted for publication once it meets all outstanding technical requirements.

Kind regards,

Mária A. Deli, M.D., Ph.D.

Academic Editor

PLOS ONE

Additional Editor Comments (optional):

Reviewers' comments:

Reviewer's Responses to Questions

**Comments to the Author**

1. If the authors have adequately addressed your comments raised in a previous round of review and you feel that this manuscript is now acceptable for publication, you may indicate that here to bypass the “Comments to the Author” section, enter your conflict of interest statement in the “Confidential to Editor” section, and submit your "Accept" recommendation.

Reviewer #2: All comments have been addressed

2. Is the manuscript technically sound, and do the data support the conclusions?

Reviewer #2: Yes

3. Has the statistical analysis been performed appropriately and rigorously? 

Reviewer #2: Yes

4. Have the authors made all data underlying the findings in their manuscript fully available?

Reviewer #2: Yes

5. Is the manuscript presented in an intelligible fashion and written in standard English?

Reviewer #2: Yes

6. Review Comments to the Author

Reviewer #2: The authors have addressed the critical concerns sufficiently in the updated version. I have no further comments.

7. PLOS authors have the option to publish the peer review history of their article (what does this mean?). If published, this will include your full peer review and any attached files.

Reviewer #2: No

---

## [Editor Report · Acceptance letter]

30 Dec 2024

PONE-D-24-13219R2 

PLOS ONE

Dear Dr. Jansen, 

I'm pleased to inform you that your manuscript has been deemed suitable for publication in PLOS ONE. Congratulations! Your manuscript is now being handed over to our production team.

Kind regards, 

on behalf of

Prof. Mária A. Deli 

Academic Editor

PLOS ONE